# Study on Plant Crushing and Soil Throwing Performance of Bionic Rotary Blades in *Cyperus esculentus* Harvesting

Hao Zhu, Dongwei Wang, Xiaoning He *, Shuqi Shang, Zhuang Zhao, Haiqing Wang, Ying Tan and Yanxin Shi

College of Mechanical and Electrical Engineering, Qingdao Agricultural University, Qingdao 266109, China; 20202204029@stu.qau.edu.cn (H.Z.); 200701031@qau.edu.cn (D.W.); sqshang@qau.edu.cn (S.S.); 20192204158@stu.qau.edu.cn (Z.Z.); 20202204018@stu.qau.edu.cn (H.W.); 20202204017@stu.qau.edu.cn (Y.T.); 20202104021@stu.qau.edu.cn (Y.S.)
* Correspondence: 201502004@qau.edu.cn

**Abstract:** Soil blocking and plant entanglement are the main challenges in *Cyperus esculentus* harvesting and excavating. The structure of the rotary blade is one of the critical factors affecting plant crushing and soil throwing performance. Inspired by the composite motion of longitudinal soil cutting and lateral soil throwing of the oriental mole cricket, a bionic rotary blade was designed with the contour curve of the excavation edge and excavation surface of its forefoot claw toe. The bionic rotary blade's mechanical and kinematic analysis revealed its cutting mechanism. A flexible plant soil, bionic, rotary blade discrete element model was developed to simulate the *Cyperus esculentus* digging process. The optimal excavation edge and excavation surface were selected by a single factor experiment, and the optimal operating parameters of the bionic rotary blade were obtained by quadratic regression orthogonal rotational combination design. The results showed that the bionic rotary blade, based on the excavation edge and excavation surface of mole cricket first claw toe, had the longest throwing distance and the largest number of broken bonds. The best combination of operating parameters of the bionic rotary blade was 11.16 mm for blade spacing, 0.66 m/s for forward speed, and 300 rpm for shaft speed. The field experiment was carried out according to the best parameters. The results showed that the bionic rotary blade's average soil throwing distance and plant crushing rate were 632.30 mm and 81.55%, respectively; thereby, not only meeting the requirements of *Cyperus esculentus* harvesting, but proving superior to the operation performance of the Chinese standard rotary blade IT245 and rotary blade with optimized cutting edge (IT245P). The results of this study can provide bionic design ideas and methods for the design of soil-cutting-based tillage machinery's soil-engaging components, such as the rotary blade and returning blade.

**Keywords:** *Gryllotalpa orientalis Burmeister*; bionic; rotary blade; *Cyperus esculentus*; discrete element method (DEM)

## 1. Introduction

In *Cyperus esculentus* harvesting, excavation is one of the most basic and critical links, which is a prerequisite to ensure the subsequent separation efficiency and harvesting quality [1–3]. At present, national standard rotary blades are often adopted to harvest *Cyperus esculentus* in most parts of China, which are usually down-cut rotary blades. When national standard rotary blades are used to harvest *Cyperus esculentus* by reverse rotary tillage, the soil throwing performance is poor. It is highly likely to cause soil blocking. In addition, due to the powerful tillering ability and developed root system of *Cyperus esculentus*, the national standard rotary blades are not effective in plant crushing and are easily entangled by the plants. These difficulties increase power consumption and seriously hinder the development of mechanized harvesting of *Cyperus esculentus* [4,5].

In natural evolution, many organisms have evolved unique physical structures for cutting objects, which offer suitable inspiration for parameter optimization of blade structures. The application of bionics has been the focus of domestic and foreign research, and the

research objects are mainly divided into two types. The first research type focuses on insect organs, or animal parts, with closed motion, such as the mouthparts of *Batocera horsfieldi* [6], the upper jaw of *Gryllidae* [7,8], and the chela of *Brachyura* [9]. When the cutting edges on both sides are closed, they can support each other. The cutting motion mode is single, and the parts responsible for cutting are generally fixed. However, due to dependence on mutual support, resulting in a limited range of applications, devices based on this type of research are commonly used in forward and reverse power stubble cutting [6]. In the cutting process, the forward and reverse disk cutter provide mutual support, and the linear velocity directions are opposite, to complete root system crushing. The second research type focuses on insect organs, or animal parts, with a unidirectional cutting motion, such as the claw toes of *Gryllotalpa orientalis Burmeister* [10–12], the claw toes of *Meles* [13], the rat toes of *Mole* [14,15], and the claws of *Canis lupus* [16], which mainly rely on sharp cutting edge and unique movement to complete the cutting operation. In current bionic research, some scholars only address the cutting edge, profile curve of static objects but do not combine the unique cutting motion to perform dynamic bionics. In terms of static and dynamic biomimicry, Yang [15] fitted the contour of the mole's forefoot with multiple toes and used a linkage mechanism to simulate the mole's soil cutting trajectory. However, due to the limitations of the linkage mechanism itself, the digging efficiency was low, and it was challenging to meet the requirements of harvesting. In addition, Xiao [12], Qi [9], Guo [14], and others changed the rotary blade edge shape to a multi-tooth structure based on the bionic principle, which had an excellent crushing effect for crops with thick stalks. However, the bionic teeth of the edge were easily entangled with grass when operating in fields with many grass stalks, resulting in lower cutting force and increased power consumption. In summary, when applying the bionic principle for cutting operations, it is necessary to select a suitable research object in combination with the operating environment, and to make an accurate contour curve fitting in combination with its unique cutting motion to achieve the expected operating effect.

Given the above problems, the *Gryllotalpa orientalis Burmeister* in the suitable soil of *Cyperus esculentus* was selected as the research object in this paper. The four claw toes were fitted with contour curves, taking into account the unique combination of longitudinal cutting and lateral throwing movements of the mole cricket. The bionic rotary blade edge curve and curved surface were designed with the contour curve of the excavation edge used for longitudinal soil cutting and the excavation surface used for lateral soil throwing. The best parameter combination of the bionic rotary blade was determined by the quadratic regression orthogonal rotational combination design. The prototype was completed, and the operational performance was tested by field experiment.

## 2. Materials and Methods

### 2.1. General Structure

The general structure of the bionic reverse rotary tiller is shown in Figure 1. The device is suspended from the rear of the tractor, which provides forward traction and rotational power for the shaft. The tillage depth can be adjusted by the depth limiting device and the shaft position. The bionic rotary blade is the most important working part of the reverse rotary tiller, with the function of cutting soil and throwing soil backward [17].

### 2.2. Design and Analysis of Bionic Rotary Blade

Traditional rotary blades are mainly used to excavate the *Cyperus esculentus* for harvesting. However, when operating in sandy loam, the loss rate of leaking beans is high, and the plant crushing of *Cyperus esculentus* is not effective. Concerning the principle of bionics, the functional biological characteristics of the mole cricket and its structural features were studied and simulated to optimize the rotary blade structure, providing innovative ideas to improve the quality of *Cyperus esculentus* excavation. The mole cricket, commonly known as the "immortal in soil", often inhabits sandy loam areas and has an excellent digging ability, capable of digging burrows up to 2–3 m long overnight. Observing the mole cricket's

appearance as shown in Figure 2, it can be noted that its forefeet are placed on both sides and point forward at 20° with the body. The forefeet are wide, and the claw toes outward bend at different angles. In digging, the exoskeleton is responsible for bearing gravity, and the muscles and joints can cooperate to achieve contraction, elongation, and twisting. The compound movement mainly includes cutting soil and throwing soil. Among these, cutting soil relies on the excavation edge of the forefoot toes, with joint *A* and joint *B* as the center of rotation to cut soil diagonally to the rear. Throwing soil mainly relies on the excavation surface of the forefoot toes, with joint *A* as the center of rotation to throw soil to both sides. When operating in the field, the bionic blade edge curve takes the role of cutting soil and plants, and the curved surface takes the role of breaking and throwing soil. It can be seen that the compound motion of the mole cricket and the bionic rotary blade, when excavating soil, are highly similar. Sandy loam is suitable soil for *Cyperus esculentus* planting, and mole crickets have a powerful ability to dig in sandy loam. The plant crushing and soil throwing performance of the bionic rotary blade were improved by using the bionic principle to fit the contour curve of the excavation edge and excavation surface in line with the forefoot claw toes of mole crickets.

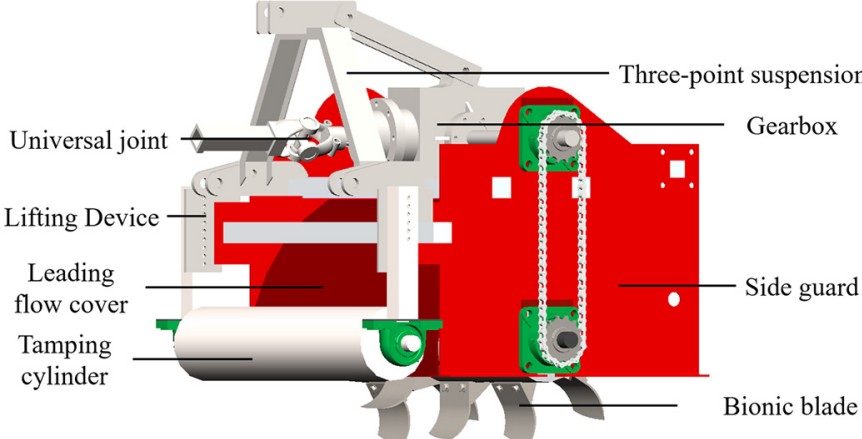

**Figure 1.** General structure of reverse rotary tiller.

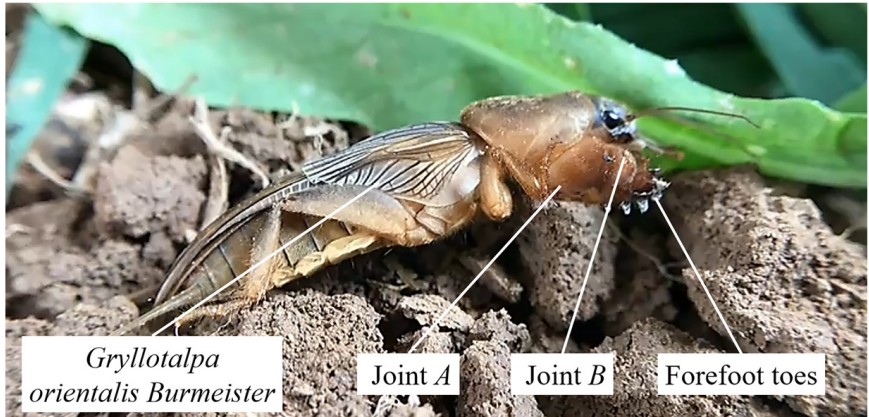

**Figure 2.** Forefoot structure of *Gryllotalpa orientalis Burmeister*.

The geometric shape of the claw toes of the mole cricket is complex and resembles a curved triangular pyramid. To obtain accurate contour curves of the excavation edge, the forefoot of the mole cricket was fixed on a foam board using a No. 0 insect pin. The upper and lower contours of the four claw toes were photographed separately by an EOS 90D camera (Canon, Tokyo, Japan), as shown in Figure 3a,b. The coordinates of the surface of the forefoot claw toes of the mole cricket were captured by a handheld 3D laser scanner (Creaform, Lévis, Quebec, Canada, type: Creaform Handyscan 700, accuracy: 0.03 mm),

and redundant data points were removed. The Imageware 13.2 software (Siemens PLM Software, Germany)was selected to smooth the point cloud, and the NURBS model was used to reconstruct the surface, as shown in Figure 3c.

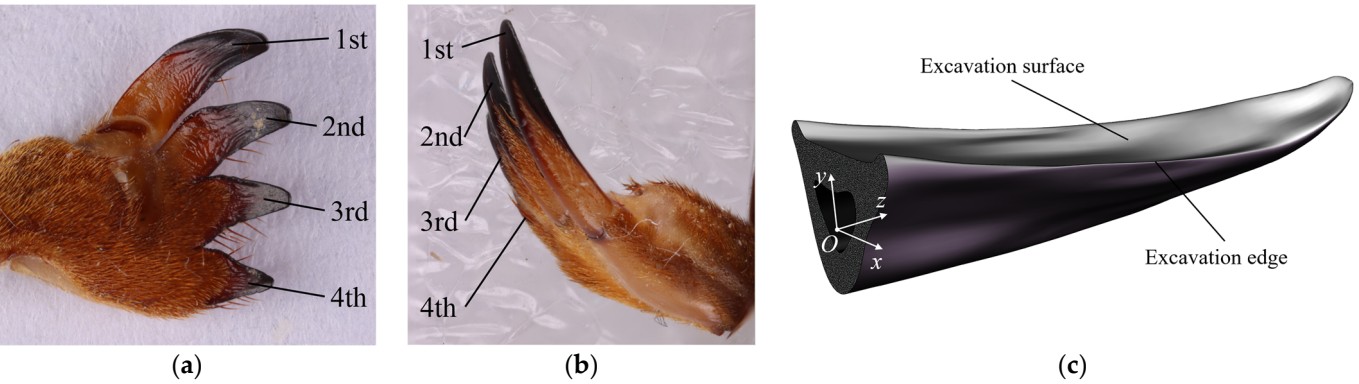

**Figure 3.** Structure and modeling of the forefoot claw toes of *Gryllotalpa orientalis Burmeister*. (**a**) front view; (**b**) side view; (**c**) inverse modeling of 1st claw toe.

### 2.2.1. Excavation Edge Contour Curve Fitting of Bionic Rotary Blade

The upper and lower contours of the four claw toes were photographed along the *y*-axis. The lower contour was the excavation edge, responsible for cutting soil during the mole cricket digging process. The point cloud data of the toe contours in the *x-z* plane were substituted into AutoCAD 2020 (Autodesk, Mill Valley, CA, USA)to obtain the toe contours' *x* and *z* coordinate values. The curves of the upper and lower contours of the forefoot claw toes of the mole cricket were fitted in the plane, based on the least-squares method, as shown in Figure 4. The general equation of the fitting equation *L* was:

$$L = a + bx + cx^2 + dx^3 + ex^4 + fx^5 + \cdots \tag{1}$$

where *a*, *b*, *c*, *d*, *e*, and *f* are polynomial coefficients. The fitting curves were drawn and scaled by CAD software.

The adjusted parameters are shown in Table 1. The determination coefficients of the fitting equations of the 1st, 2nd, 3rd and 4th claw toes were 0.9996, 0.9985, 0.9910 and 0.9909, respectively. The greater the coefficient of determination of the fitting equation, the better the fitting degree. The fitting degree was over 0.99 when the fitting equation was a quartic polynomial, indicating that the fitting curve could well reflect the excavation edge contour of the mole cricket claw toe.

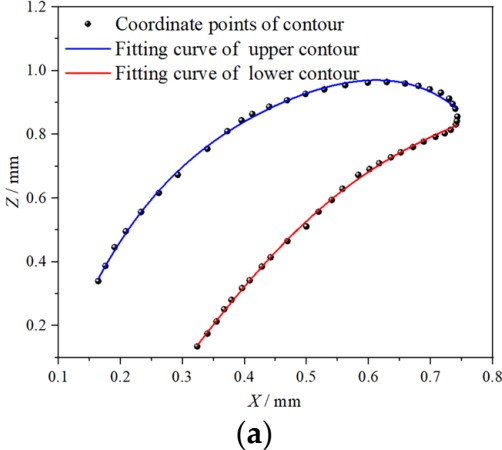

**(a)**

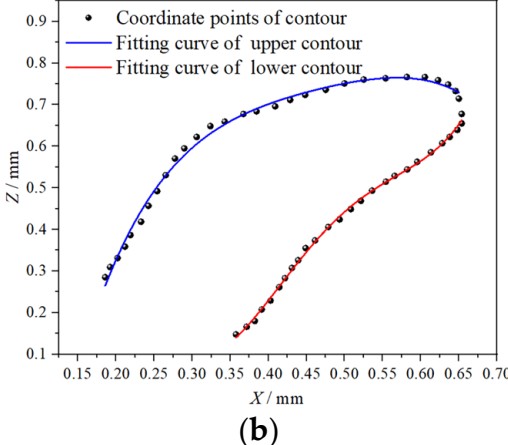

**(b)**

**Figure 4.** *Cont*.

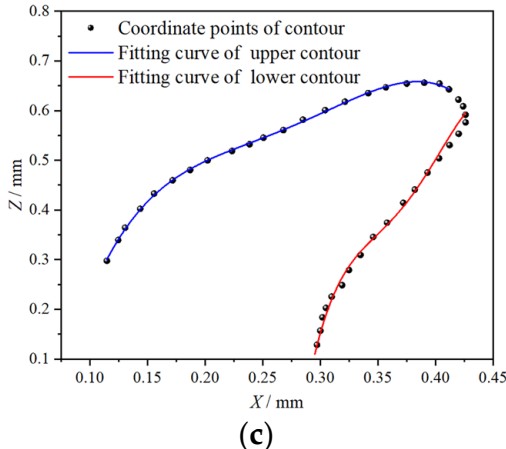

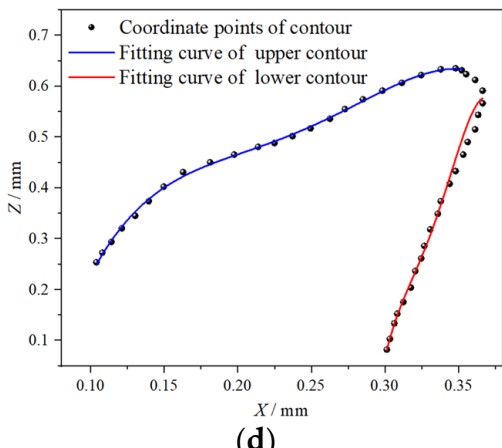

(c)

(d)

**Figure 4.** The fitting curve of upper and lower contours of the excavation edge in the *x-z* plane. (**a**) 1st claw toe; (**b**) 2nd claw toe; (**c**) 3rd claw toe; (**d**) 4th claw toe.

**Table 1.** Coefficients of the fitting curve equation for the toe excavation edge of the mole cricket.

| Coefficients | 1st Claw Toe | 2nd Claw Toe | 3rd Claw Toe | 4th Claw Toe |
|:---:|:---:|:---:|:---:|:---:|
| $a$ | −0.5581 | 8.557 | −0.9531 | 824.9 |
| $b$ | 0.775 | −77.55 | 20.33 | 9979 |
| $c$ | 7.97 | 254.5 | −110 | −45,250 |
| $d$ | −13.65 | −353.5 | 273.2 | 91,120 |
| $e$ | 6.599 | 179.6 | −250.2 | −68,700 |
| $R^2$ | 0.9996 | 0.9985 | 0.9910 | 0.9909 |

### 2.2.2. Excavation Surface Contour Curve Fitting of Bionic Rotary Blade

The upper and lower contours of the four claw toes were photographed along the *x*-axis direction, where the lower contour was the excavation surface, responsible for throwing soil during the mole cricket digging process. The claw toe upper and lower contour curves in the *y-z* plane were fitted, as shown in Figure 5. The general equation of the fitting equation *S* was

$$S = a + bx + cx^2 + dx^3 + ex^4 + fx^5 + \cdots \tag{2}$$

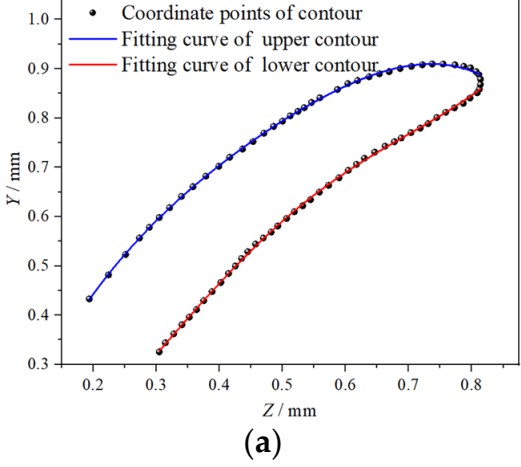

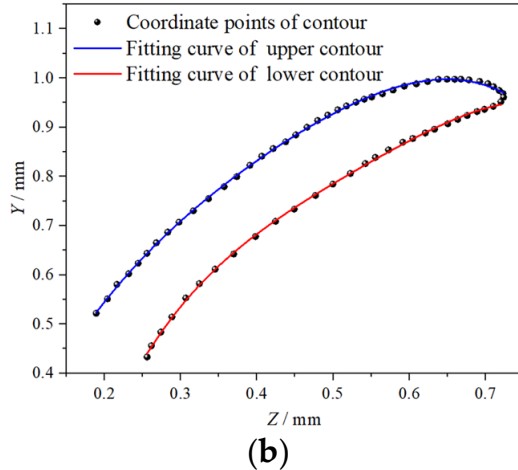

(a)

(b)

**Figure 5.** *Cont.*

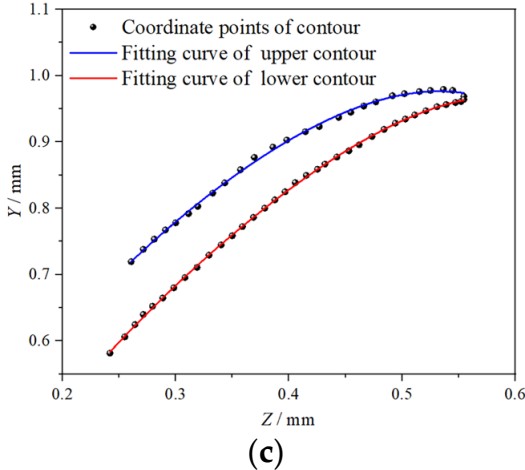
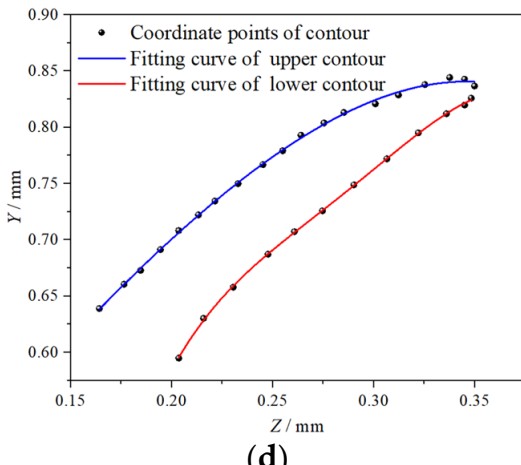

**(c)**                                                                                                                    **(d)**

**Figure 5.** The fitting curve of upper and lower contours of the excavation surface in the *y-z* plane. (**a**) 1st claw toe; (**b**) 2nd claw toe; (**c**) 3rd claw toe; (**d**) 4th claw toe.

The adjusted parameters are shown in Table 2. The determination coefficients of the fitting equations of the 1st, 2nd, 3rd and 4th claw toes were 0.9997, 0.9996, 0.9998 and 0.9997, respectively. The fitting degree was over 0.99 when the fitting equation was a quartic polynomial, indicating that the fitting curve could well reflect the excavation surface contour of the mole cricket claw toe.

**Table 2.** Coefficients of the fitting curve equation for the toe excavation surface of the mole cricket.

| Coefficients | 1st Claw Toe | 2nd Claw Toe | 3rd Claw Toe | 4th Claw Toe |
|:---:|:---:|:---:|:---:|:---:|
| *a* | 0.1142 | −1.032 | −0.07473 | −5.031 |
| *b* | −0.9333 | 10.65 | 4.085 | 77.29 |
| *c* | 8.84 | 26.44 | −8.222 | −401 |
| *d* | −13.42 | −32.42 | 13.01 | 938.9 |
| *e* | 6.579 | −15.2 | −9.705 | −821.6 |
| $R^2$ | 0.9997 | 0.9996 | 0.9998 | 0.9997 |

2.2.3. Parameter Design of Bionic Rotary Blade

The bionic rotary blade radius was the sum of the shaft radius, the spacing from the shaft to the ground, and the depth of the bionic rotary blade into the soil [18]. The larger the diameter of the bionic rotary blade, the better the cutting performance, but the higher the power and the torque applied to the shaft [19]. The root depth of *Cyperus esculentus* in Henan *Cyperus esculentus* planting base was obtained as 115 ± 5 mm. To reduce the loss of leaking beans, the tillage depth of the bionic rotary blade was taken as 130 mm. To ensure the shaft strength and prevent interference, the shaft radius and the distance from the shaft to the ground were determined as 150 and 100 mm, respectively, so the radius of the bionic rotary blade *R* was 380 mm.

Since the bionic rotary blade edge was curved, one side of the blade was low, and the other side was high. In order to ensure tillage depth stability and reduce the loss of leakage during *Cyperus esculentus* harvesting, the high point height of the blade $h_1$ was taken as 130 mm. The low point height of the blade $h_2$ was calculated by Equation (1), and the transverse projection length *l* of the curved surface was calculated by Equation (2). The mole cricket forefoot is a combination of four claw toes with different spacing between claw toes. In order to investigate the influence of the spacing between adjacent excavation surfaces on the soil throwing performance, the spacing between adjacent blades was taken as 0 mm, 10 mm, 20 mm, and 30 mm, taking into account the environment and the requirements of the *Cyperus esculentus* harvesting operation. The general structure of bionic rotary blade is shown in Figure 6.The general structure parameters are shown in Table 3.

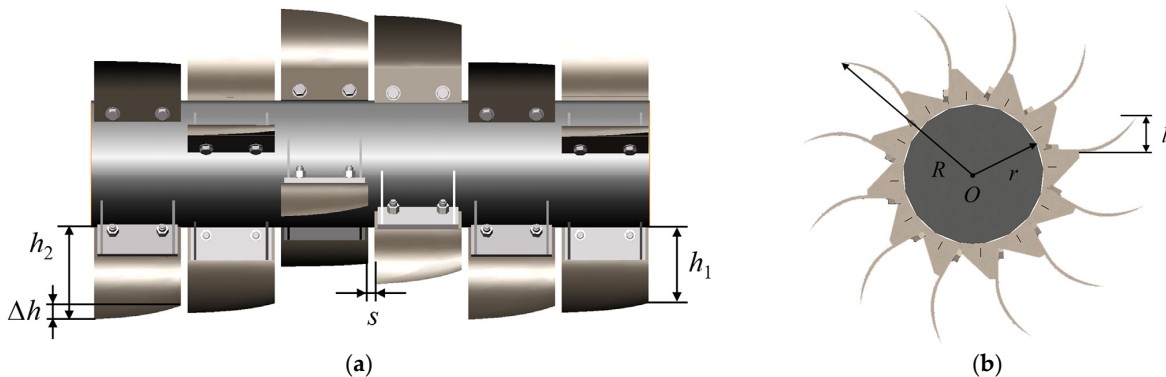

**Figure 6.** General structure of bionic rotary blade. (**a**) front view; (**b**) side view, where *s* is the blade spacing (mm); $h_1$ is the distance from the high point of the cutting edge to the shaft (mm); $h_2$ is the distance from the low point of the cutting edge to the shaft (mm); $\Delta h$ is the distance between the high point and the low point of the cutting edge (mm); *r* is the shaft radius (mm); *R* is the maximum turning radius of the bionic rotary blade (mm); *L* is the projection length of bionic rotary blade curved surface (mm).

**Table 3.** Structural parameters of the bionic rotary blade corresponding to different claw toes.

| Claw Toe | $h_1$ | $h_2$ | $\Delta h$ | $l$ |
|----------|-------|-------|------------|-----|
| 1st | 130.00 | 117.64 | 12.36 | 24.74 |
| 2nd | 130.00 | 111.60 | 18.40 | 17.39 |
| 3rd | 130.00 | 120.53 | 9.47 | 13.46 |
| 4th | 130.00 | 123.11 | 6.89 | 8.38 |

*2.3. Cutting Mechanics Analysis*

2.3.1. Force Analysis of the Soil Unit

During the cutting process of the bionic rotary blade, the soil was impacted by the blades. Under the positive pressure of the blades, the soil was squeezed and compacted. When the size of the force on the soil unit was sufficient to destroy the friction and cohesion inside the soil, the soil unit was separated from the soil and thrown out by the bionic rotary blade. When cutting soil, the force mainly came from the friction between the blade's two sides and the soil, the friction between the blade and the ditch bottom, and the soil resistance, as shown in Figure 7. The direction of soil shear stress was the same as the linear velocity direction of the bionic rotary blade, and the direction of soil resistance was the opposite of the linear velocity direction.

The force system balance equation of the bionic rotary was:

$$\begin{cases} N_1 + N_2 - p = 0 \\ P_0 + T - F - f_1 - f_2 = 0 \end{cases} \tag{3}$$

According to the equation defined by Magalhaes [20], the soil resistance *F* was:

$$F = \left( \rho_d g h^2 N_{\rho_d} + ch N_c c_a N_{c_a} \right) \tag{4}$$

where $\rho_d$ is the soil bulk density (kg/m³); *g* is the acceleration of gravity(m/s²); *h* is the tillage depth (m); *c* is the soil cohesion (Pa); $c_a$ is the soil adhesion (Pa); $N_{\rho d}$, $N_c$ and $N_{ca}$ are the soil resistance coefficients of the bionic rotary blade. The shear stress of the bionic rotary blade cutting soil was obtained by:

$$T = \rho_d g h^2 N_{\rho_d} + ch N_c c_a N_{c_a} + f_1 + \mu(p - N_1) - P_0 \tag{5}$$

where $\mu$ is the friction coefficient between the bionic rotary blade and the soil.

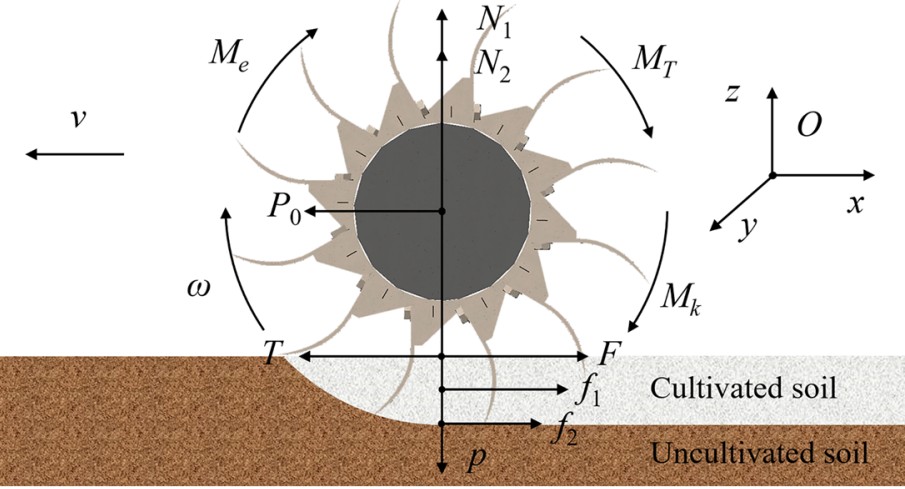

**Figure 7.** Force analysis of soil cutting process by bionic rotary blade. where $\omega$ is the angular velocity of the bionic rotary blade, rad/s; $N_1$ is the support force of the soil on the bionic rotary blade, N; $N_2$ is the support force of the shaft on the bionic rotary blade, N; $F$ is the resistance of the bionic rotary blade to cut the soil, N; $f_1$ is the friction force between the two sides of the bionic rotary blade and the soil, N; $f_2$ is the friction force between the bionic rotary blade and the ditch bottom, N; $T$ is the shear stress of the bionic rotary blade to cut the soil, N; $p$ is the downward pressure on the bionic rotary blade, N; $P$ is the traction force, N; $M_e$ is the torque provided by the tractor to the bionic rotary blade, N·m; $M_T$ is the torque of the bionic rotary blade to cut the soil, N·m; $M_k$ is the torque of the bionic rotary blade to cut the root system, N·m.

The moments of the bionic rotary blade were divided into soil cutting moments and root cutting moments. To obtain the cutting shear stress of the bionic rotary blade on the root system, the moment analysis was performed, and the moment analysis is shown in Figure 7. The moment balance equation of the bionic rotary was:

$$M_e = M_T + M_k \tag{6}$$

The cutting shear stress of the bionic rotary blade on the root system was given by:

$$F_k = \frac{M_e - TL_k}{l_k} \tag{7}$$

where $L_k$ is the shear stress arm of the bionic rotary blade cutting the soil, mm; $l_k$ is the shear stress arm of the bionic rotary blade cutting the root system, mm.

### 2.3.2. Critical Conditions for Shear Damage in *Cyperus esculentus* Plants

When the bionic rotary blades cut the *Cyperus esculentus* plants, some leaves slide out along the blade edge, because the leaves are located at the surface and have no fixed restraint; the roots are bound by the soil and broken under the cutting action of the bionic rotary blade. *Cyperus esculentus* plants have strong tillering power and well-developed root systems, closely connected to the soil to form agglomerates. Root fracture is crucial to breaking the agglomerates of *Cyperus esculentus*. Therefore, the analysis of the critical conditions for shear damage of *Cyperus esculentus* plants focused on the root system. In order to obtain the critical conditions for shear damage of the *Cyperus esculentus* plant by the bionic rotary blade, force analysis of the process of cutting the *Cyperus esculentus* plant was performed, as shown in Figure 8.

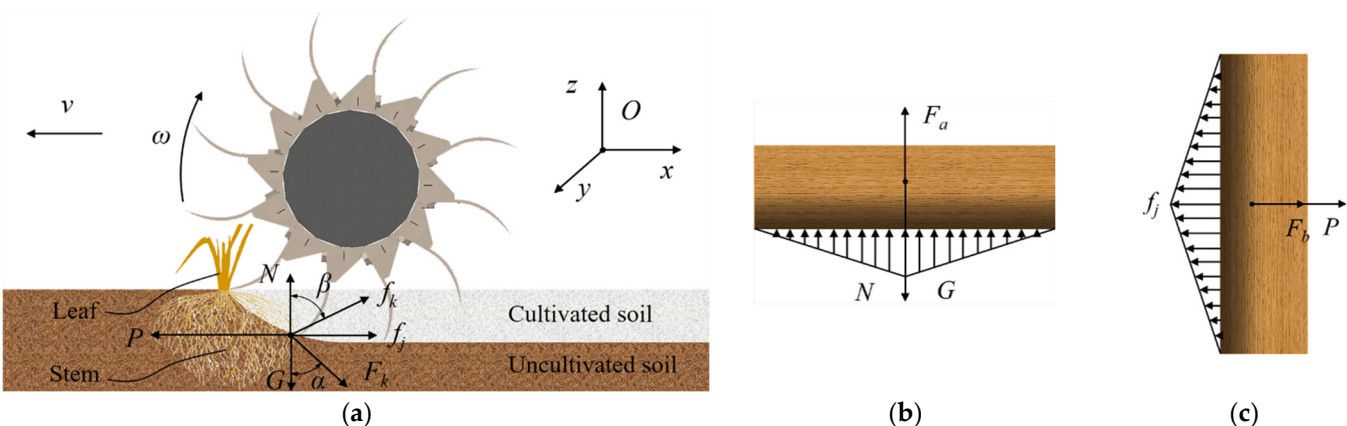

**Figure 8.** Force analysis of root cutting process. (**a**) *x-z* plane; (**b**) *y-z* plane; (**c**) *x-y* plane. where $N$ is the support force of the ground on the root system (N); $G$ is the gravity of the root system (N); $P$ is the thrust of the machine on the root system (N); $f_k$ is the frictional force of the bionic rotary blade on the root system, (N); $f_j$ is the frictional force of the root system with the ground, (N); $F_a$ is the force on the root system in the normal direction due to the bionic rotary blade cutting, (N); $F_b$ is the force on the root system in the horizontal direction due to the bionic rotary blade cutting (N); $\alpha$ is the angle between $F_k$ and the normal direction (°); $\beta$ is the angle between $f_k$ and the normal direction (°).

Since the direction of shear stress in the root system of *Cyperus esculentus* plants was the direction of linear velocity at the contact point, the root system was subjected to external forces in both the normal and horizontal directions. From Figure 8, the external force on the root system under the action of the bionic rotary blade was:

$$\begin{cases} F_a = \cos \alpha F_k - \cos \beta f_k \\ F_b = \sin \alpha F_k + \sin \beta f_k \end{cases} \tag{8}$$

The static equilibrium equation was obtained by:

$$\begin{cases} F_a - G + N = 0 \\ F_b - f_j + P = 0 \end{cases} \tag{9}$$

When the shear stress in the normal and horizontal directions was greater than the allowable stress of the root system, the root system was cut off. The critical conditions for shear damage in the normal and horizontal directions of the root system of the *Cyperus esculentus* plant can be obtained by:

$$\tau_1 = \frac{1}{A_0}(N + F_a) = \frac{1}{A_0}(Nl + \cos \alpha M_e - T \cos \alpha L + \cos \beta f_k l) \tag{10}$$

$$\tau_2 = \frac{1}{A_0}(F_b + P) = \frac{1}{A_0}(Pl + \sin \alpha M_e - T \sin \alpha L + \sin \beta f_k l) \tag{11}$$

where $A_0$ is the cross-section area of the root system of the *Cyperus esculentus* plant, mm$^2$. From Equations (10) and (11), the cutting performance of the bionic rotary blade was related to the torque provided by the tractor to the bionic rotary blade roller and the shear stress of cutting soil. According to theoretical mechanics, the shaft torque is related to the shaft speed, i.e., the cutting performance of the bionic rotary blade is affected by the shaft speed.

*2.4. Kinematic Analysis*

2.4.1. Kinematic Analysis of Bionic Rotary Blade

Kinematic analysis of the bionic rotary blade was carried out to determine the kinematic parameters affecting operational performance. When digging *Cyperus esculentus*, the bionic blade rotated in a circular motion with the shaft as the center, while moving straight

under the tractor's traction. With the center of rotation of the rotary blade as the coordinate origin to establish a coordinate system, as shown in Figure 9, the kinematic of the bionic rotary blade tips were analyzed. The trajectory equation of the low point $A$ and the high point $B$ of the bionic rotary blade was:

$$\begin{cases} x_1 = vt + R_A \cos(\theta_1 - \omega t) \\ y_1 = R_A \sin(\theta_1 - \omega t) \end{cases} \tag{12}$$

$$\begin{cases} x_2 = vt + R_B \cos(\theta_2 + \omega t) \\ y_2 = R_B \sin(\theta_2 + \omega t) \end{cases} \tag{13}$$

where, $t$ is the operation time of the bionic rotary blade, s; $R_A$ is the gyration radius of the blade tip $A$, mm; $R_B$ is the gyration radius of the blade tip $B$, mm.

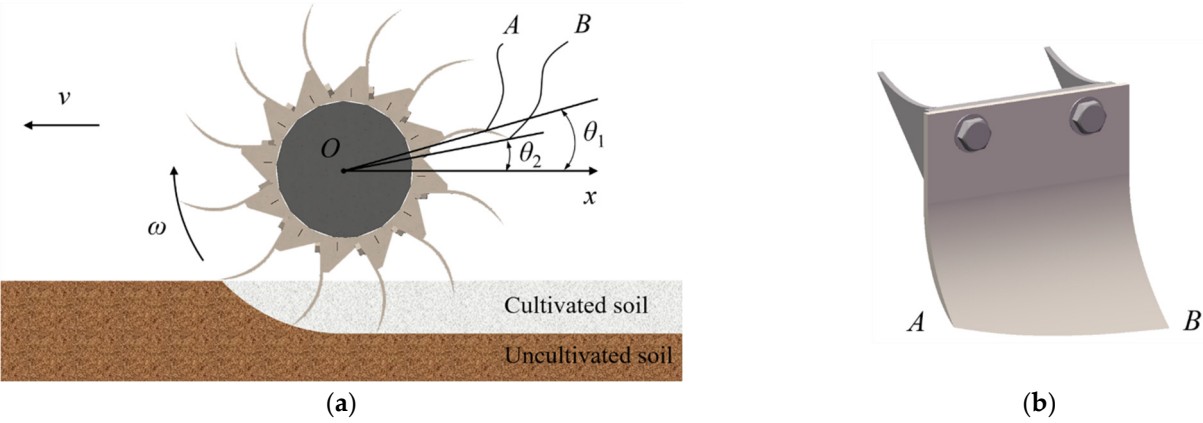

(**a**)  (**b**)

**Figure 9.** Bionic rotary blade motion analysis. (**a**) side view of motion analysis; (**b**) axonometric drawing of bionic rotary blade. where $A$ is the low point of the bionic rotary blade; $B$ is the high point of the bionic rotary blade; $\theta_1$ is the angle between point $A$ and the positive direction of the $x$-axis (°); $\theta_2$ is the angle between point $B$ and the positive direction of the $x$-axis (°).

When the bionic rotary blade completed a circular motion, the motion equation of motion blade tips $A$ and $B$ was:

$$\begin{cases} \theta_1 + 2k\pi = \theta_1 - 2\pi\omega l/v \\ \theta_2 + 2k\pi = \theta_2 - 2\pi\omega l/v \end{cases} \tag{14}$$

where $l$ is the length of the experimental site (m); $k$ is an integer. From Equation (14), the root cutting frequency was related to the shaft and forward speeds. The root cutting frequency was related to the cutting performance of the bionic rotary blade. Thus, the shaft speed and the forward speed were the motion parameters that affected the plant crushing performance of the bionic rotary blade.

### 2.4.2. Kinematic Analysis of the Soil Unit

When the blade comes out of the trench, the soil unit is brought up by the blade and subject to gravity, centrifugal force, and the force between the soil when air resistance is ignored, as shown in Figure 10.

The condition for the soil unit to leave the tool was that the outward force was greater than its friction and support components:

$$\mu(G + F_c \cos\alpha)\sin\alpha - N_x \leqslant F_c \sin\alpha \tag{15}$$

where $\mu$ is the friction coefficient between the bionic rotary blade and the soil unit.

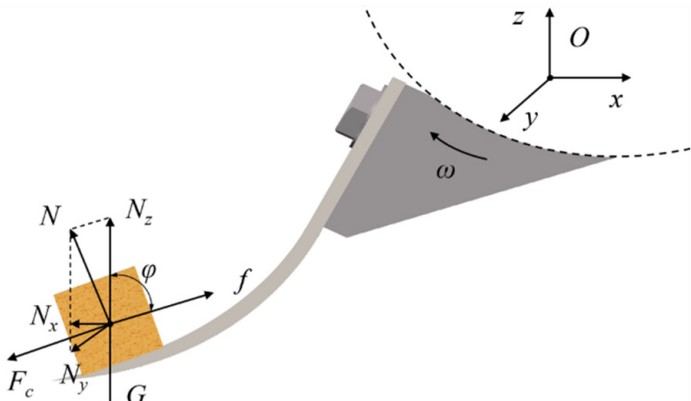

**Figure 10.** Force diagram of the soil unit. where $G$ is the gravity of the soil unit (N); $f$ is the friction force between the soil unit and the bionic rotary blade (N); $N$ is the support force of the bionic rotary blade on the soil unit (N); $N_x$, $N_y$, and $N_z$ are the component forces of the support force $N$ in the $x$, $y$ and $z$ axes, respectively (N); $F_c$ is the centrifugal force on the soil unit (N); $\varphi$ is the rotation angle of the bionic rotary blade when the soil unit leaves the bionic rotary blade (°).

The velocity direction of the soil unit leaving the blade is determined by the curvature of the bionic rotary blade surface [21]. After the soil unit leaves the blade, its motion direction changes due to air resistance, the collision between soil units, and the collision between soil units and the cover. The main influencing factors are the size and direction of the velocity of the soil unit and the position and surface shape of the dividing plate. When the force between the soil and the air resistance is ignored, after the soil unit leaves the tool, it is only acted on by gravity and will make a parabolic motion with the velocity size and direction when it leaves the blade. The maximum horizontal distance $s_{max}$ and the maximum height $h_{max}$ of the parabolic motion of the soil unit, without considering the influence of the cover [21], were calculated by:

$$\begin{cases} x = v_0 t \cos \beta \\ y = v_0 t \sin \beta - g t^2 / 2 \end{cases} \tag{16}$$

$$s_{max} = v_0^2 \sin(2\beta) / g \tag{17}$$

$$h_{max} = v_0^2 \sin^2(2\beta) / 2g \tag{18}$$

where, $x$ is the $x$ coordinate of the soil unit (m); $y$ is the $y$ coordinate of the soil unit (m); $v_0$ is the velocity of the soil unit when it leaves the bionic rotary blade (m/s); $t$ is the time when the soil unit moves to the coordinate $(x,y)$ (s); $\beta$ is the throwing angle of the soil unit (°); $s_{max}$ is the maximum distance of the soil unit in the horizontal direction (m); $h_{max}$ is the maximum height of the soil unit in the vertical direction (m).

The soil throwing performance of the bionic rotary blade was related to the centrifugal force applied to the soil unit and the initial speed of leaving the bionic rotary blade, as shown by the throwing conditions and the motion state after throwing of the soil unit. The centrifugal force and the initial speed were mainly influenced by the bionic rotary blade speed and the forward speed of the implement [21], so the shaft speed and the forward speed were the kinematic parameters that affected the throwing performance of the bionic rotary blade.

*2.5. Discrete Element Simulation*

The discrete element model of the "plant soil bionic rotary blade" was established by EDEM software, and the material of the bionic rotary blade was 65 Mn. The contact parameters of the model were set as shown in Table 4. The contact model of soil was the Hertz-Mindlin (no-slip) model with a particle radius of 2 mm. The contact model of the *Cyperus esculentus* plant was Hertz-Mindlin with a bonding model with a contact radius of

3.35 mm for *Cyperus esculentus* leaves and 2.25 mm for *Cyperus esculentus* roots, and 3 mm for bonding [22], as shown in Figure 11. The soil bin size was 1000 mm × 550 mm × 360 mm, and the soil particles were 2 mm in diameter. According to the plant spacing and row spacing of the *Cyperus esculentus* planting pattern, eight plant generation plants with length × width × height of 280 mm × 200 mm × 200 mm were established in the soil bin model, and the plant spacing and row spacing were set to 200 mm and 300 mm, respectively. The particle bonding time was set to 0.5 s. The motion parameters of the bionic rotary blade were set according to the experimental scheme, and the discrete element simulation process is shown in Figure 12.

**Table 4.** Parameters of discrete element model.

| | Parameter | | Value |
|---|---|---|---|
| Intrinsic parameter | Soil | Density/(kg·m$^{-3}$) | 2679 |
| | | Shear Modulus/(Pa) | $1.2 \times 10^5$ |
| | | Poisson's Ratio | 0.36 |
| | Plant | Density/(kg·m$^{-3}$) | 470 |
| | | Shear Modulus/(Pa) | $1.7 \times 10^6$ |
| | | Poisson's Ratio | 0.40 |
| | 65 Mn | Density/(kg·m$^{-3}$) | 7850 |
| | | Shear Modulus/(Pa) | $7.9 \times 10^{10}$ |
| | | Poisson's Ratio | 0.30 |
| Contact Parameter | Soil-Soil | Restitution Coefficient | 0.55 |
| | | Static Friction Coefficient | 0.15 |
| | | Rolling friction Coefficient | 0.43 |
| | Plant-Plant | Restitution Coefficient | 0.34 |
| | | Static Friction Coefficient | 0.08 |
| | | Rolling friction Coefficient | 0.35 |
| | Soil-Plant | Restitution Coefficient | 0.48 |
| | | Static Friction Coefficient | 0.05 |
| | | Rolling friction Coefficient | 0.32 |
| | Soil-65 Mn | Restitution Coefficient | 0.52 |
| | | Static Friction Coefficient | 0.12 |
| | | Rolling friction Coefficient | 0.20 |
| | Plant-65 Mn | Restitution Coefficient | 0.33 |
| | | Static Friction Coefficient | 0.10 |
| | | Rolling friction Coefficient | 0.30 |
| Contact Model | | Normal Contact Stiffness/(N·m$^{-3}$) | $1.723 \times 10^8$ |
| | | Tangential Contact Stiffness/(N·m$^{-3}$) | $9.075 \times 10^7$ |
| | | Normal Critical Stress/(Pa) | $2.215 \times 10^5$ |
| | | Tangential Critical Stress/(Pa) | $2.215 \times 10^5$ |

In order to analyze the effects of the bionic rotary blade edge curve and curved surface on the excavation quality of *Cyperus esculentus*, two sets of single-factor experiments were designed to analyze the effects of each parameter on the soil throwing distance and the number of broken bonds. The experiment parameters are shown in Table 5.

Single-factor experiments were conducted to determine the appropriate edge curve and curved surface for the bionic rotary blade harvesting *Cyperus esculentus*. The results of the mechanical and kinematic analysis showed that the operating quality of the bionic rotary blade was influenced by the blade spacing, forward speed, and shaft speed. A quadratic regression orthogonal rotational combination design was conducted using Design-Expert software for blade spacing, forward speed, and shaft speed to obtain the bionic rotary tillage device's optimal structural and kinematic parameters. The test factors and levels are shown in Table 6.

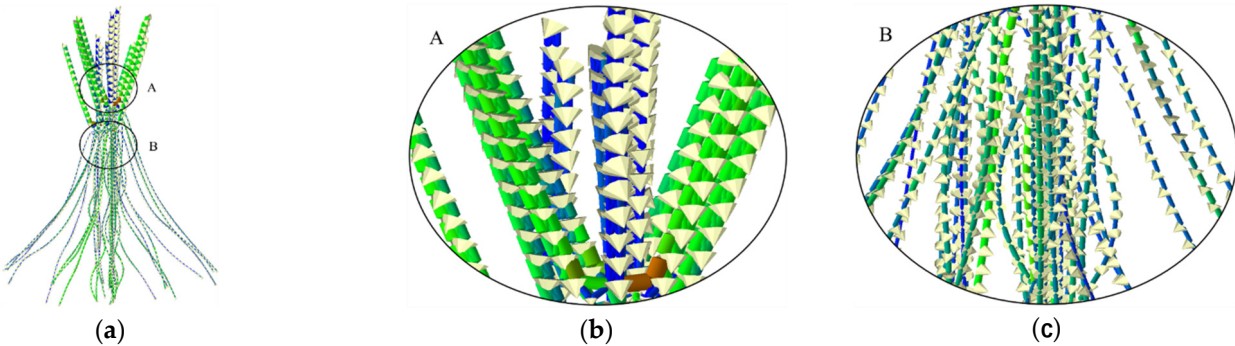

**Figure 11.** Flexible model of *Cyperus esculentus* plants. (**a**) whole plant; (**b**) bonds between leaf particles; (**c**) bonds between root particles.

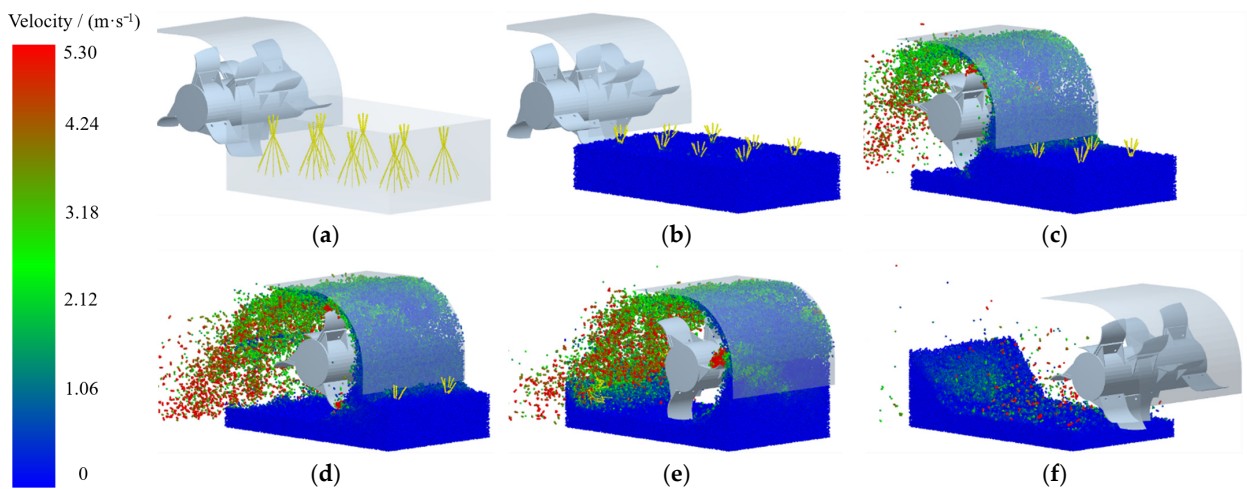

**Figure 12.** Simulation process of reverse rotary tillage for *Cyperus esculentus* harvesting. (**a**) 0.6 s; (**b**) 1.5 s; (**c**) 2.5 s; (**d**) 5 s; (**e**) 7 s; (**f**) 9 s.

**Table 5.** Single-factor experiments design.

| Group | Test Factor | | Constant Factor | | |
|---|---|---|---|---|---|
| | Edge Curve | Curved Surface | Blade Spacing /(mm) | Forward Speed /(m·s$^{-1}$) | Shaft Speed /(r·min$^{-1}$) |
| 1 | Excavation edge of 1st/2nd/3rd/4th claw toe | Excavation surface of 2nd claw toe | 20 | 0.56 | 250 |
| 2 | Excavation edge of 2nd claw toe | Excavation surface of 1st/2nd/3rd/4th claw toe | 20 | 0.56 | 250 |

**Table 6.** Test factors and levels.

| Code | Blade Spacing/(mm) | Forward Speed/(m·s$^{-1}$) | Shaft Speed/(r·min$^{-1}$) |
|---|---|---|---|
| 1.682 | 30 | 0.83 | 300 |
| 1 | 26 | 0.73 | 280 |
| 0 | 20 | 0.56 | 250 |
| −1 | 14 | 0.39 | 220 |
| −1.682 | 10 | 0.28 | 200 |
| $\Delta j$ | 6 | 0.17 | 30 |

*2.6. Evaluation Index of Cyperus esculentus Harvesting*

2.6.1. Soil Throwing Performance

There are tens of thousands of soil particles in the simulation test, and it is difficult to calculate the soil throwing distance as the particle trajectories are interlaced and complicated during the bionic rotary blade throwing process. Therefore, a longitudinal belt of soil particles was set at 300 mm intervals. Each soil particle belt contained ten tracer particles dyed black to observe the dynamic movement process of cutting soil during the operation of the bionic rotary blade, as shown in Figure 13.

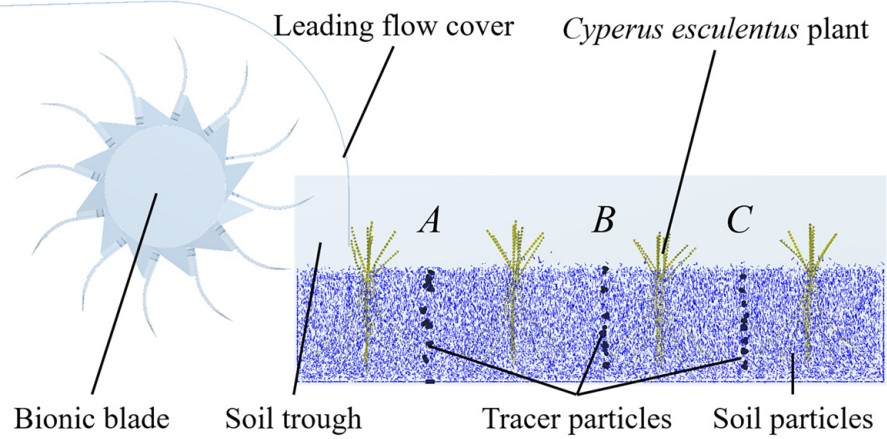

**Figure 13.** Tracer soil particle setup, where *A*, *B* and *C* are three soil particle bands.

In the process of soil throwing by the bionic rotary blade, the trajectory of soil particles is shown in Figure 14. Take soil particle belt *A* as an example. The tracer particles were pushed and squeezed by the bionic rotary blade, displaced for a distance in the forward direction, then moved upward obliquely under the throwing action of the bionic rotary blade and performed a parabolic motion along with the cover to the rear. The starting position of the single soil particle belt was consistent, and the landing position of each soil particle was recorded. The distance between the starting position and the landing position was the soil throwing distance *S*. The larger the soil throwing distance, the better the throwing performance of the bionic rotary blade.

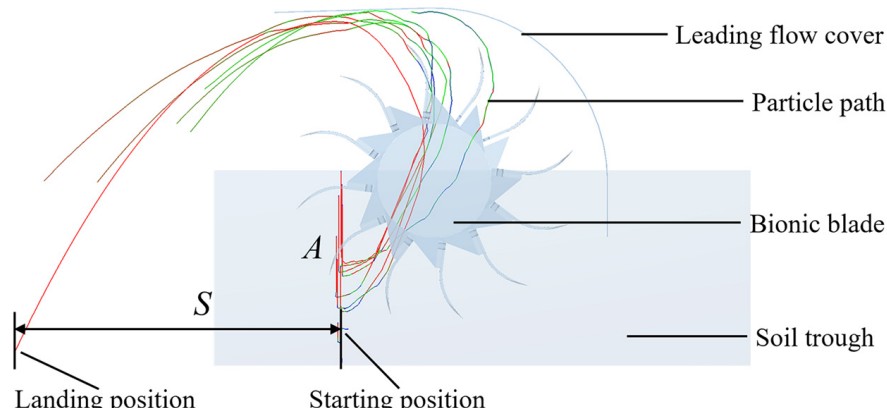

**Figure 14.** Motion trajectory of tracer soil particles, where *A* is the soil particle band; *S* is the soil particle throwing distance (mm).

2.6.2. Plant Crushing Performance

The soil particles and *Cyperus esculentus* plant particles were concealed for observation, and only the bonds between the *Cyperus esculentus* plant particles were retained. As the bionic rotary blade rotated and cut, the interparticle bonding band broke after the *Cyperus esculentus* plant model reached its force limit, i.e., the *Cyperus esculentus*

plant was shredded by the blade [20,22,23]. The higher the number of broken bonds of *Cyperus esculentus* plants, the better the plant crushing performance of the bionic rotary blade. The breakage process of the *Cyperus esculentus* plant particles bonds is shown in Figure 15.

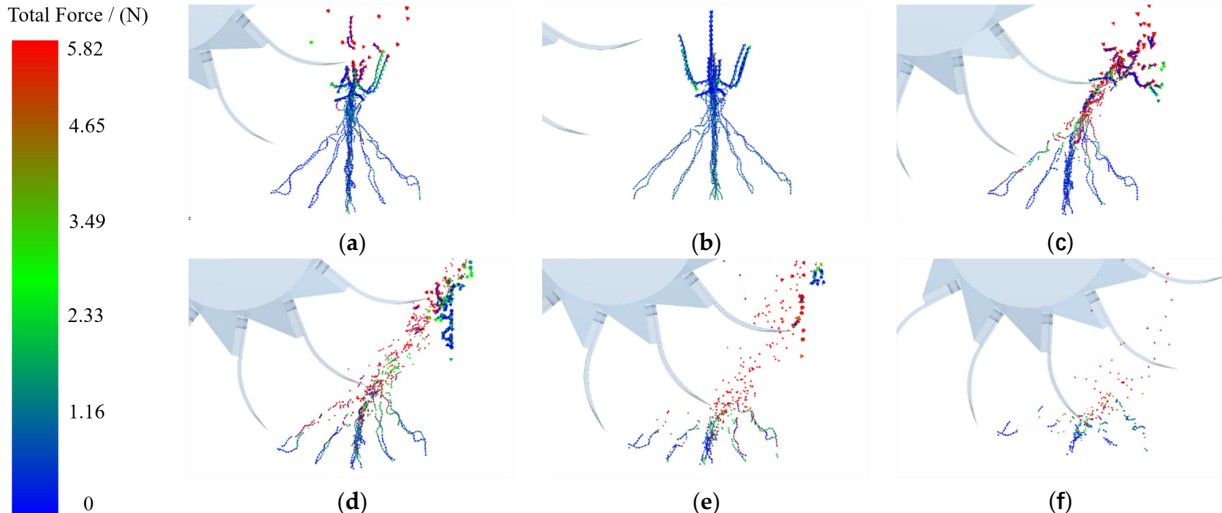

**Figure 15.** The crushing process of *Cyperus esculentus* plants. (**a**) 2.3 s; (**b**) 2.5 s; (**c**) 2.8 s; (**d**) 3 s; (**e**) 3.2 s; (**f**) 3.5 s.

### 2.7. Field Experiment

To test the effect of the bionic rotary blade for *Cyperus esculentus* excavation, a field experiment was conducted on 15 November 2021, at the *Cyperus esculentus* planting base in Minquan County, Shangqiu City, Henan Province (115°18′ E, 34°31′ N), as shown in Figure 16. The soil type of the selected test plots was sandy loam with a soil moisture content of $15.36 \pm 1.56\%$, soil density of $2650 \pm 50$ kg·m$^{-3}$, and soil firmness of $425 \pm 50$ kPa. The row spacing and plant spacing of *Cyperus esculentus* were $200 \pm 15$ mm and $150 \pm 10$ mm, respectively. The operating conditions of the selected test plots were the same as in [24–26]. The test apparatus was a Dongfeng DF604 tractor (Changzhou Dongfeng Agricultural Machinery Group Co., Ltd., Changzhou, China, with a matching power of 60 hp) and a bionic reverse rotary tiller. The data collection instruments were SN-101-2QB electric constant temperature blast dryer (Shanghai Shangyi Co., Ltd., Shanghai, China), JC-XM-D electric soil relative density meter (Qingdao Juchuang Digital Co., Ltd., Qingdao, China), and JC-JSD-01 soil compactness tester (Qingdao Juchuang Digital Co., Ltd., Qingdao, China). Other auxiliary tools included a tape measure, stainless steel ruler, soil extraction cutting ring, and shovel.

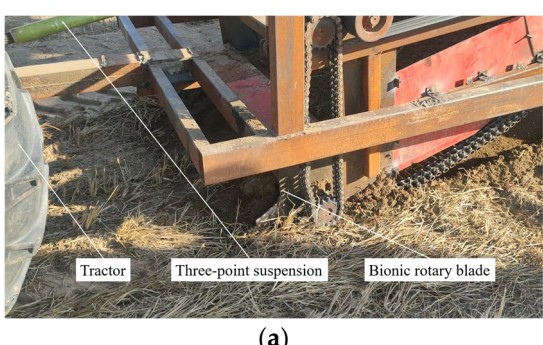

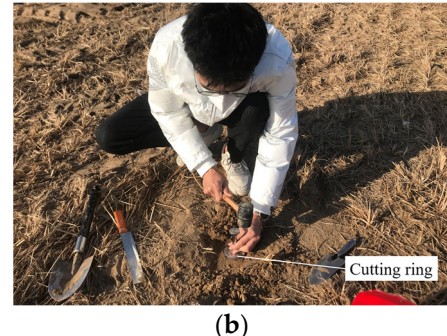

**Figure 16.** Field experiment. (**a**) test device; (**b**) soil sample collection.

Before the field experiment, 10 test points were selected equidistantly along the forward direction of the machine with the starting point as the origin, and each test point was buried with ten black plastic blocks in advance. Ten color blocks were distributed equidistantly along the vertical direction in the depth range of 0–130 mm [27,28]. The starting and the landing positions of the color blocks were recorded, and the distance between the starting and the landing positions was the soil throwing distance *S*.

After the field experiment was completed, ten 1 m × 1 m testing areas were randomly selected in the operation area. Since there is no relevant standard for the crushing effect of *Cyperus esculentus* plants, the crushing rate of *Cyperus esculentus* plants in this test was calculated by referring to the requirements of the straw crushing qualified rate in GB/T24675.62009 "Conservation Tillage Machinery Straw Crushing and Returning Machine". The leaves and roots of the *Cyperus esculentus* plant, which were not broken, the length of the root system, which was more than 130 mm, and the length of leaves, which were more than 90 mm, were not to be cut off. The plant crushing rate *y* of *Cyperus esculentus* was calculated by:

$$y = \sum_{i=1}^{10} \frac{M_{ai} - M_{bi}}{M_{ai}} \times 100\% \tag{19}$$

where $M_{ai}$ is the total mass of *Cyperus esculentus* plants in the testing area (kg); $M_{bi}$ is the mass of uncut *Cyperus esculentus* plants in the testing area (kg); *i* is the serial number of the testing area.

The power consumption of rotary tiller was mainly related to shaft torque and forward resistance, which can be calculated by [29]:

$$W = \frac{nM}{9550} + Fv_m \tag{20}$$

where *W* is the power consumption (kW); *n* is the shaft speed (r/min); *M* is the shaft torque (N·m); *F* is the forward resistance (N); $v_m$ is the forward speed (m/s).

## 3. Results and Analysis

### 3.1. Single Factor Experiment

3.1.1. Effect of Bionic Rotary Blade Edge Curve on Operation Quality

The relationship curve of the bionic rotary blade edge curve on the operation quality is shown in Figure 17. When the bionic rotary blade edge curve changed from 1st claw toe to 4th claw toe, the number of broken bonds decreased rapidly from 2336 to 1635, and the curve showed a linearly decreasing trend. This was because the excavation edge curve from 1st claw toe to 4th claw toe gradually tended to straighten. The curvature of the excavation edge curve had a positive effect on crushing plants, and the plant crushing performance of the 1st claw toe excavation edge was the best. When the claw toe changed from 1st to 4th, the soil throwing distance fluctuated in the range 642–654 mm. The minimum value was at the 3rd claw toe, and the soil throwing distance was 642 mm. The maximum value was at the 2nd claw toe, the soil throwing distance was 654 mm. Soil throwing performance was improved by only 1.87%. It can be seen that the bionic rotary blade edge curve had little effect on the soil throwing performance. This was consistent with the results of Liu et al. [27,28] on submerged soil rotary tillage. The soil throwing performance was mainly affected by the throwing surface, while the change of the edge curve had no significant effect on the throwing surface. In China, the national standard rotary blade IT245 is often used to harvest *Cyperus esculentus*. In the previous phase, simulation and field trials were conducted for the performance of rotary blade IT245 for harvesting *Cyperus esculentus* [29]. When the rotary blade IT245 and rotary blade with optimized cutting edge (IT245P) were simulated, the number of broken bonds were 1575 and 1762, respectively. The parameter settings of the simulation test were kept constant in this study, and the number of broken bonds of the bionic rotary blade ranged from 1635 to 2336. Comparing the results of the studies, it can be seen that the bionic rotary blade, based

on different claw toes' edge curves had different degrees of improvement in crushing *Cyperus esculentus* plants, compared to the rotary blades IT245 and IT245P. Compared to rotary blades IT245 and IT245P, the rotary blade with the 1st claw toe excavation edge curve had a 48.32% and 32.58% increase in the number of broken bonds, respectively. In summary, the 1st claw toe excavation edge curve was selected to design the bionic rotary blade edge curve when conducting the quadratic regression orthogonal rotational combination design.

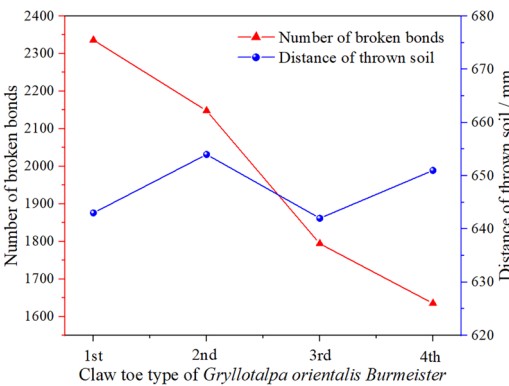

**Figure 17.** Relationship curves between bionic rotary blade edge curve and operation quality.

### 3.1.2. Effect of Bionic Rotary Blade Curved Surface on Operation Quality

The relationship curve of the bionic rotary blade curved surface on the operation quality is shown in Figure 18. When the bionic rotary blade curved surface was changed from 1st claw toe excavation surface to 4th claw toe excavation surface, the number of broken bonds tended to decrease linearly in the range 1848~2358, so the plant crushing effect of the 1st claw toe excavation surface was the best. The parameter settings of the simulation test were kept constant in this study. Comparing the results of the studies [29], it can be seen that the bionic rotary blade, based on different claw toes' curved surfaces, had significant improvement in crushing *Cyperus esculentus* plants, compared to the rotary blades IT245 and IT245P. Compared to rotary blades IT245 and IT245P, the rotary blade with the 1st claw toe excavation surface curve had a 49.71% and 33.83% increase in the number of broken bonds, respectively. When the claw toe changed from 1st to 3rd, the soil throwing distance decreased linearly from 511 mm to 645 mm. When changing from the excavation face of 3rd claw toe to 4th claw toe, the soil throwing distance decreased slowly in the range of 489–511 mm, so the throwing effect of the excavation face of the 1st claw toe was the best. This was consistent with the findings of Wang et al. [30] on rice straw returns. Compared with the flat surface, the curved surface had an accelerating effect on the soil and, thus, had good soil throwing performance. The 1st claw toe excavation surface curve was selected to design the bionic rotary blade curved surface when conducting the quadratic regression orthogonal rotational combination design.

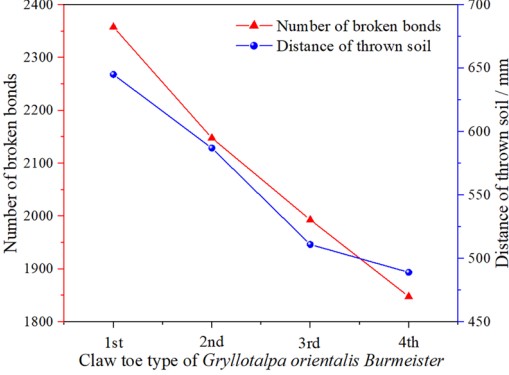

**Figure 18.** Relationship curves between bionic rotary blade curved surface and operation quality.

### 3.2. Results of Quadratic Regression Orthogonal Rotational Combination Design

The experimental design scheme and results are shown in Table 7.

**Table 7.** Experimental protocol and results.

| No. | Test Factor | | | Evaluation Index | |
|---|---|---|---|---|---|
| | Blade Spacing $X_1$ | Forward Speed $X_2$ | Shaft Speed $X_3$ | Soil Throwing Distance $Y_1$/mm | Number of Broken Bonds $Y_2$ |
| 1 | −1 | −1 | −1 | 623.01 | 2233 |
| 2 | 1 | −1 | −1 | 613.65 | 2201 |
| 3 | −1 | 1 | −1 | 660.18 | 2196 |
| 4 | 1 | 1 | −1 | 623.68 | 2189 |
| 5 | −1 | −1 | 1 | 649.01 | 2438 |
| 6 | 1 | −1 | 1 | 653.68 | 2398 |
| 7 | −1 | 1 | 1 | 676.45 | 2386 |
| 8 | 1 | 1 | 1 | 673.57 | 2334 |
| 9 | −1.682 | 0 | 0 | 652.74 | 2398 |
| 10 | 1.682 | 0 | 0 | 637.30 | 2297 |
| 11 | 0 | −1.682 | 0 | 633.70 | 2406 |
| 12 | 0 | 1.682 | 0 | 657.73 | 2265 |
| 13 | 0 | 0 | −1.682 | 626.74 | 2174 |
| 14 | 0 | 0 | 1.682 | 667.81 | 2458 |
| 15 | 0 | 0 | 0 | 645.44 | 2347 |
| 16 | 0 | 0 | 0 | 642.61 | 2349 |
| 17 | 0 | 0 | 0 | 644.98 | 2355 |
| 18 | 0 | 0 | 0 | 642.85 | 2354 |
| 10 | 0 | 0 | 0 | 644.05 | 2361 |
| 20 | 0 | 0 | 0 | 644.10 | 2349 |
| 21 | 0 | 0 | 0 | 648.18 | 2357 |
| 22 | 0 | 0 | 0 | 648.56 | 2366 |
| 23 | 0 | 0 | 0 | 641.33 | 2336 |

As shown in Table 8, the *p* values of all models were less than 0.01, indicating that the models were highly significant and could predict the soil throwing distance and the number of broken bonds well. The larger the coefficient of determination $R^2$ (0~1), the better the fit of the model. The coefficients of determination $R^2$ for the models of soil throwing distance and number of broken bonds were 0.9564 and 0.9517, respectively, indicating that the predicted values of the regression equations were in good agreement with the actual values. The smaller the coefficient of variation CV (0–1), the better the reliability of the test [31]. The CV values of soil throwing distance and number of broken bonds were 0.66% and 1.00%, respectively, indicating that the simulated test had good reliability.

Removing non-significant influences, the quadratic polynomial regression equation for the soil throwing distance and the number of broken bonds was:

$$\begin{cases} Y_1 = 644.67 - 5.13X_1 + 9.88X_2 + 14.74\ X_3 - 4.34X_1X_2 + 5.96X_1X_3 \\ Y_2 = 2353.42 - 22.03X_1 - 29.45X_2 + 88.94\ X_3 - 13.31X_2{}^2 - 20.21X_3{}^2 \end{cases} \quad (21)$$

From Equation (21), soil throwing distance was linearly and positively correlated with the forward speed $X_2$ and shaft speed $X_3$. The reason was that the soil throwing distance was directly proportional to the initial speed of soil leaving the rotary blade, and the initial speed was directly proportional to the forward speed and shaft speed. Therefore, the soil throwing distance increased with increase of forward speed and shaft speed. This is consistent with the results of Yang's study [22]. The number of broken bonds was quadratically related to the forward speed $X_2$ and shaft speed $X_3$, which indicated an optimal combination of parameters of blade spacing $X_1$, forward speed $X_2$ and shaft speed $X_3$ to make the maximum number of broken bonds. The optimal operation parameters were obtained through 3D response surface analysis. The number of broken bonds was

linearly and positively correlated with the shaft speed $X_3$, indicating that the number of broken bonds increased with increase of shaft speed. The reason was that when the shaft speed increased, the thickness of the soil unit cut by the rotary blade decreased, so the *Cyperus esculentus* plants in the soil could be sufficiently crushed, i.e., the number of broken bonds increased. This is consistent with the results of Wang's study [30].

**Table 8.** Analysis of variance of polynomial models for quadratic regression orthogonal rotational combination design.

| Source | Soil Throwing Distance | | | | Number of Broken Bonds | | | |
|---|---|---|---|---|---|---|---|---|
| | Sum of Squares | Mean Square | F Value | p-Value | Sum of Squares | Mean Square | F Value | p-Value |
| Model | 5114.31 | 568.26 | 31.66 | <0.01 ** | 137,917.50 | 15,324.16 | 28.47 | <0.01 ** |
| $X_1$ | 359.17 | 359.17 | 20.01 | <0.01 ** | 6627.98 | 6627.98 | 12.31 | <0.01 ** |
| $X_2$ | 1333.38 | 1333.38 | 74.28 | <0.01 ** | 11,840.99 | 11,840.99 | 21.99 | <0.01 ** |
| $X_3$ | 2965.99 | 2965.99 | 165.23 | <0.01 ** | 108,028.09 | 108,028.09 | 200.70 | <0.01 ** |
| $X_1X_2$ | 150.42 | 150.42 | 8.38 | 0.0125 * | 21.12 | 21.13 | 0.04 | 0.8460 |
| $X_1X_3$ | 283.82 | 283.82 | 15.81 | <0.01 ** | 351.12 | 351.13 | 0.65 | 0.4338 |
| $X_2X_3$ | 0.002 | 0.002 | 0.0002 | 0.9915 | 561.12 | 561.13 | 1.04 | 0.3259 |
| $X_1{}^2$ | 0.91 | 0.91 | 0.051 | 0.8254 | 1307.13 | 1307.14 | 2.43 | 0.1431 |
| $X_2{}^2$ | 3.74 | 3.74 | 0.21 | 0.6558 | 2816.01 | 2816.01 | 5.23 | 0.0396 * |
| $X_3{}^2$ | 17.07 | 17.07 | 0.95 | 0.3473 | 6487.94 | 6487.95 | 12.05 | <0.01 ** |
| Residual | 233.36 | 17.95 | | | 6997.10 | 538.23 | | |
| Lack of Fit | 185.80 | 37.16 | 6.25 | 0.0119 | 6387.10 | 1277.42 | 16.75 | 0.0005 |
| Pure Error | 47.56 | 5.95 | | | 609.99 | 76.24 | | |
| | $R^2 = 0.9564$; C.V. = 0.66% | | | | $R^2 = 0.9517$; C.V. = 1.00% | | | |

Note: $0.01 < p < 0.05$ (significant, *); $0.001 < p < 0.01$ (highly significant, **).

From Table 8, the effects of blade spacing $X_1$, forward speed $X_2$ and shaft speed $X_3$ on the soil throwing distance $Y_1$ were highly significant ($p < 0.01$); the interaction term of blade spacing and shaft speed $X_1X_3$ had a highly significant effect on the soil throwing distance $Y_1$ ($p < 0.01$); the interaction term of blade spacing and forward speed $X_1X_2$ had a significant effect on the soil throwing distance $Y_1$ ($p < 0.05$). Other test factors did not have significant effects on the soil throwing distance. Ignoring the non-significant factors, and comparing the $F$ values, the order of the degree of influence of each factor on the soil throwing distance was $X_3 > X_2 > X_1 > X_1X_3 > X_1X_2$. This was consistent with the findings of Liu [32] on soil throwing motion. The soil throwing distance was proportional to the initial velocity of soil leaving the blade, and the shaft speed enhanced the initial velocity of soil more significantly. Blade spacing $X_1$, forward speed $X_2$, and shaft speed $X_3$ had highly significant effects on the number of broken bonds $Y_2$ ($p < 0.01$); shaft speed interaction term $X_3{}^2$ had highly significant effects on the number of broken bonds $Y_2$ ($p < 0.01$); forward speed interaction term $X_2{}^2$ had significant effects on the number of broken bonds $Y_2$ ($p < 0.05$). The other test factors did not significantly affect the number of broken bonds. Neglecting the non-significant factors, and comparing the $F$ values, the order of the degree of influence of each factor on the number of broken bonds was $X_3 > X_2 > X_1 > X_3{}^2 > X_2{}^2$. This was consistent with the findings of a study conducted by Lu [33] on maize root stubble cutting.

### 3.3. Response Surface Analysis

From the response surface of soil throwing distance Figure 19, the contours of blade spacing $X_1$ and forward speed $X_2$ produced approximately concentric circles, indicating that the interaction between them was not apparent. With the increase of forward speed, the soil throwing distance showed a slow-growth trend; with the increase of blade spacing, the soil throwing distance changed little and showed a slight decreasing trend. The maximum throwing distance was 670 mm when the forward speed was 0.83 m/s, and the blade spacing was 10 mm. The soil throwing distance increased linearly with the increase of shaft speed. With the increase of blade spacing, the change in soil throwing distance was small, and when the shaft speed was at a high level, the blade spacing had almost no effect on the soil throwing distance. The maximum soil throwing distance was 660 mm when the

shaft speed was 300 rpm. This was consistent with the findings of Liu et al. [27,28,34] that the counter-rotating throwing effect of the bionic rotary blade on the soil was enhanced as the initial velocity of the soil leaving the bionic rotary blade increased with the increase of shaft speed. This was consistent with the findings of Zhu and Ren et al. [21,35], where the soil throwing distance was proportional to the initial velocity, and, thus, the soil throwing distance increased. The interaction between the forward speed $X_2$ and the shaft speed $X_3$ was highly significant. When the shaft speed was at a low level, the soil throwing distance was little affected by the forward speed; when the shaft speed was at a high level, the soil throwing distance increased linearly with the increase of the forward speed. When the forward speed was low, the soil throwing distance increased slowly with the increase of forward speed; when the forward speed was high, the soil throwing distance increased linearly with the increase of forward speed. Therefore, the value of blade spacing had little effect on the soil throwing distance. Increasing the shaft speed and forward speed within a reasonable range had a facilitating effect on improving the throwing performance of the bionic rotary blade excavation, which was consistent with the findings of Liu et al. [33,36,37] regarding the movement of the throwing body.

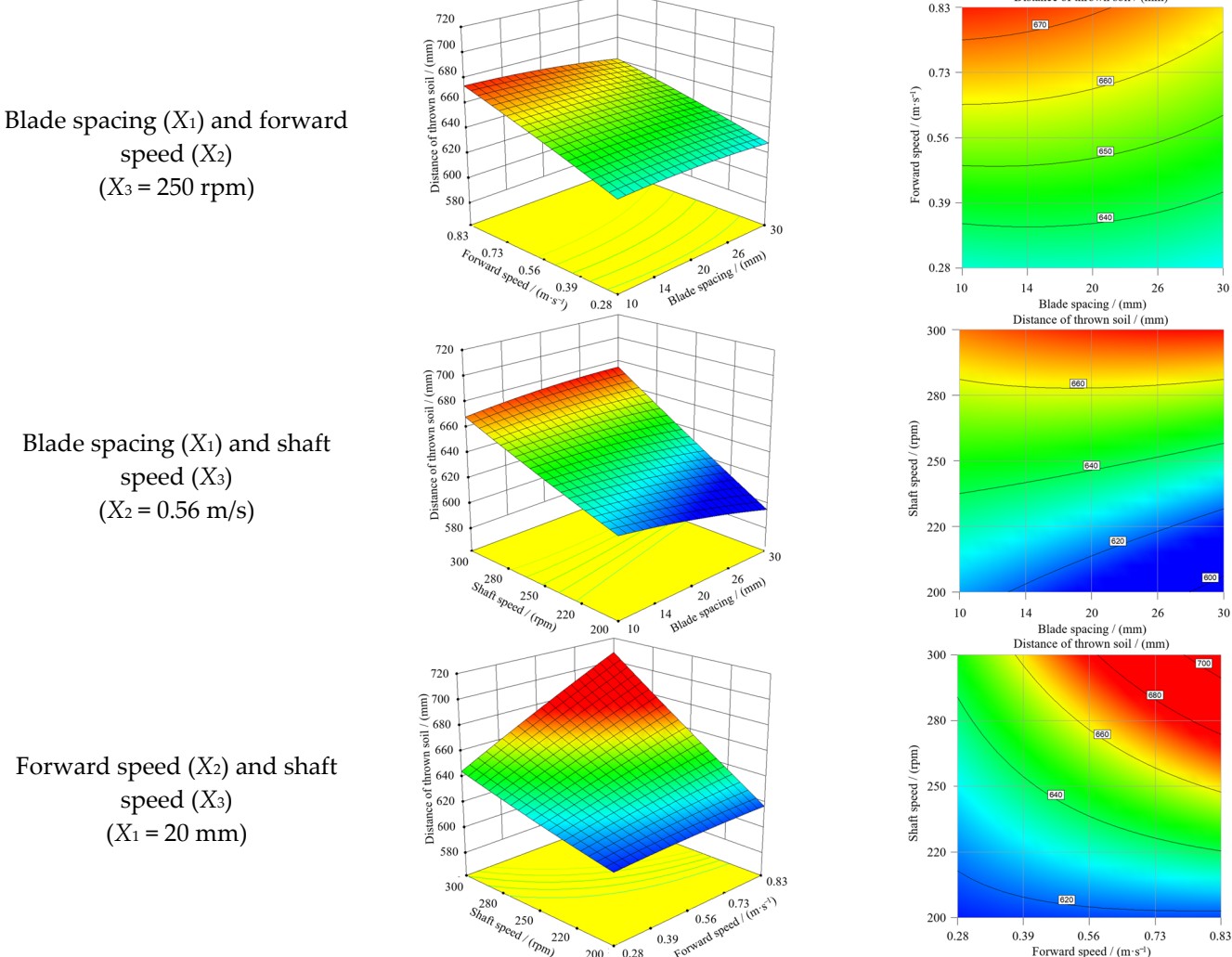

**Figure 19.** 3D response surface plot and 2D contour plot of soil throwing distance.

From Figure 20, the response surface plot of the number of broken bonds showed that the number of broken bonds was inversely proportional to the blade spacing and the forward speed, and the maximum number of broken bonds was 2450 when the blade spacing was 10–14 mm, and the forward speed was 0.28–0.39 m/s. This was consistent

with the results of Zhang et al. [38,39]. The forward speed was proportional to the cutting pitch, so the greater the forward speed, the thicker the soil unit, resulting in abundant *Cyperus esculentus* roots being uncut and thrown backward, having failed to be sufficiently broken. When the thickness of the soil unit exceeded the critical point, the resistance of the rotary cutter reached its peak and was only related to the conditions of the rotary cutter and soil parameters and was little affected by the forward speed. Hence, the number of broken bonds changed little when the forward speed was at a high level [30,40]. When the shaft speed was at a low level, the number of broken bonds was little affected by the blade spacing. When the shaft speed was at a high level, the number of broken bonds tended to decrease with increase of blade spacing, and the rate of decrease increased with the increase of forward speed. The maximum number of broken bonds was 2400 when the blade spacing was 10–20 mm, and the shaft speed was 280–300 rpm, which was consistent with the results of Zhang and Wang et al. [38,39,41]. The shaft speed was inversely proportional to the cutting pitch, so as the shaft speed increased, the cutting pitch decreased, and the *Cyperus esculentus* agglomerates were sufficiently broken. When the blade spacing was large, it led to a large gap between adjacent soil units when the bionic blades cut soil alternately, and the soil and plants in the gap were thrown backward under the action of the bionic blades before they were broken, leading to a decrease in the number of broken bonds [6]. When the shaft speed was at a low level, the number of broken bonds slowly increased in the interval of 0.28–0.56 m/s and decreased in the interval of 0.56–0.83 m/s with the increase of forward speed; when the shaft speed was at a high level, the number of broken bonds showed a linearly decreasing trend with the increase of forward speed. The number of broken bonds was overall proportional to the shaft speed. This was consistent with the findings of Zhu et al. [6] when they conducted experiments on corn stubble cutting devices. The maximum number of broken bonds was 2500 at 0.28–0.45 m/s for forward speed and 280–300 rpm for shaft speed. Therefore, within a reasonable range, decreasing the blade spacing and forward speed, and increasing the shaft speed, had a facilitating effect on improving the plant crushing performance when digging with the bionic rotary blade.

### 3.4. Parameter Optimization

In order to obtain the optimal parameter combinations for *Cyperus esculentus* harvesting with the bionic rotary blade, the regression model of the soil throwing distance and the number of broken bonds was solved by using the optimization module Design-Expert software. The constraints are shown in Equation (22). The objective function was solved optimally, and the optimal combination of parameters was 11.16 mm for blade spacing, 0.66 m/s for forward speed, and 300 rpm for shaft speed. Under these conditions, the soil throwing distance and the broken bonds were 676.45 mm and 2440, respectively. When the rotary blade IT245 and rotary blade IT245P were simulated, the number of broken bonds were 1575 and 1762, respectively [29]. Under the same simulation test conditions, the bionic rotary blade had a 35.45% and 27.79% increase in the number of broken bonds compared to rotary blades IT245 and IT245P.

$$\begin{cases} \max Y_1(X_1, X_2, X_3) \\ \max Y_2(X_1, X_2, X_3) \\ \text{s.t} \begin{cases} 10 \leqslant X_1 \leqslant 30 \\ 0.28 \leqslant X_2 \leqslant 0.83 \\ 200 \leqslant X_3 \leqslant 300 \end{cases} \end{cases} \quad (22)$$

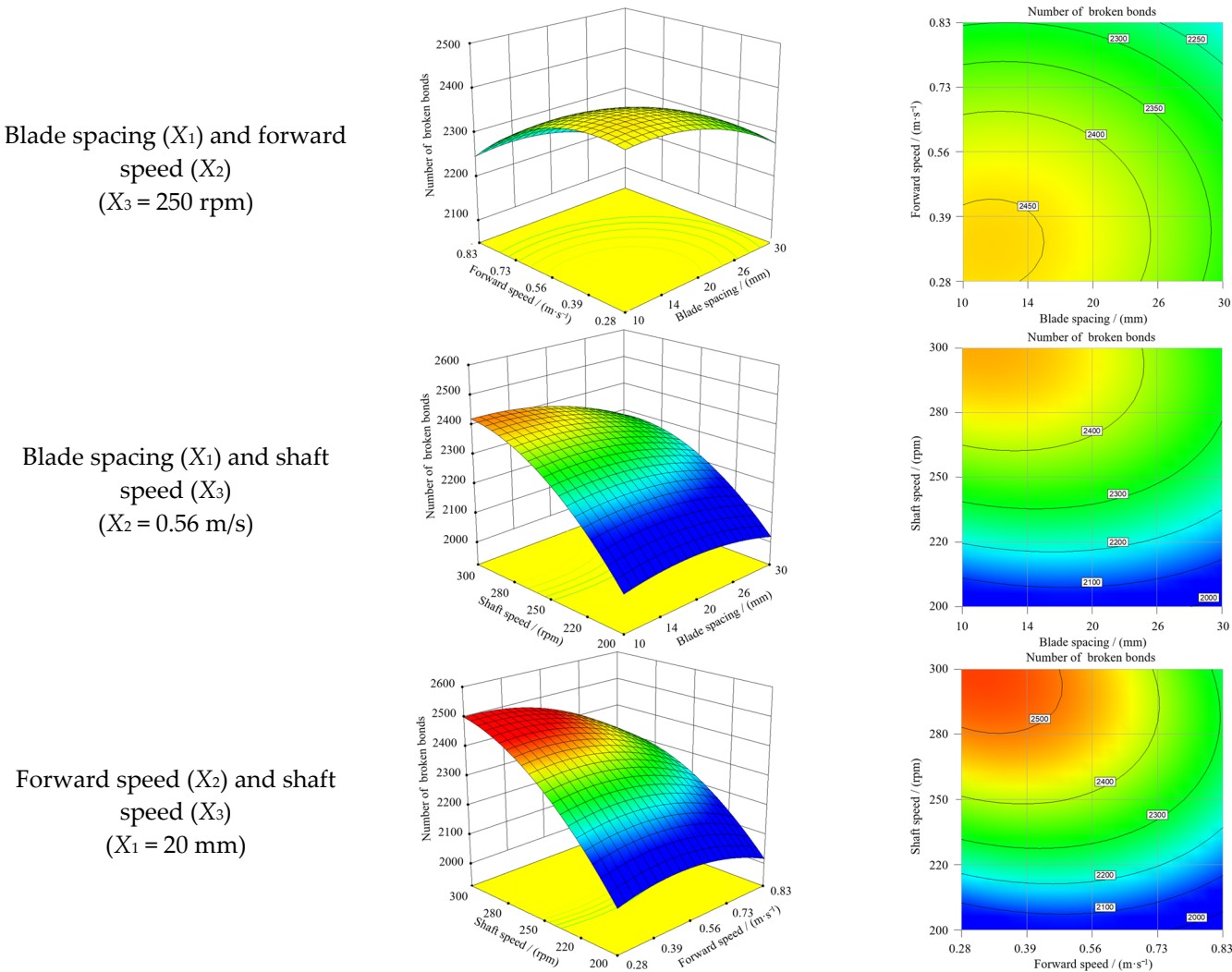

**Figure 20.** 3D response surface plot and 2D contour plot of number of broken bonds.

*3.5. Results and Analysis of Field Experiment*

During the field experiment, the bionic rotary blade operated continuously and normally, as shown in Figure 21a. From Figure 21b, the average soil throwing distance of the bionic rotary blade was 632.30 mm. Compared with the simulation test results, the average soil throwing distance in the field experiment was reduced by 6.53%. The reason may be that the actual field soil contained a small number of impurities, such as stones. However, the effect of impurities was not considered in the simulation test, so the actual soil throwing performance was slightly reduced. There was no apparent soil blocking during machine advancement in front of the device. However, the soil layer would interfere with the side plates because of the small distance from the ground. A small amount of soil would accumulate on both sides of the device, and the problem of soil blocking could be further solved by adjusting the baffle plate's installation angle and the distance from the ground. From Figure 22a, after the operation of the bionic rotary blade, the *Cyperus esculentus* plants were cut and broken into short stalks. No apparent clumping mixture appeared in the soil. From Figure 22b, the average plant crushing rate of *Cyperus esculentus* was 81.55%. In the field trial, the power consumption of the bionic rotary blade was 37.48 kW. In the previous study, the power consumption of the rotary blades IT245 and IT245P were 43.90 kW and 38.82 kW, respectively, when harvesting *Cyperus esculentus* [29]. The bionic rotary blade exhibited a 14.62% and 3.45% decrease in power consumption, compared to the IT245 IT245P rotary blades, respectively. In conclusion, the bionic rotary blade had good soil throwing

and plant crushing performance, which could effectively solve the main problems during *Cyperus esculentus* harvesting, such as severe soil blocking and low plant crushing rate.

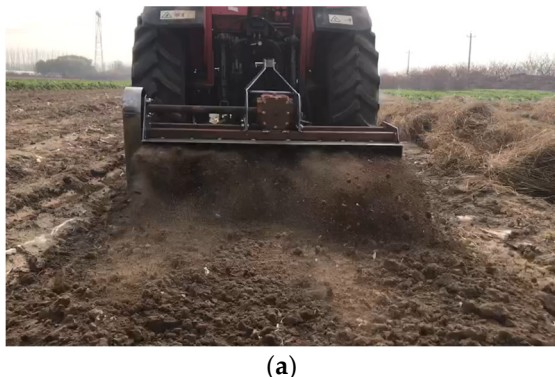

(**a**)

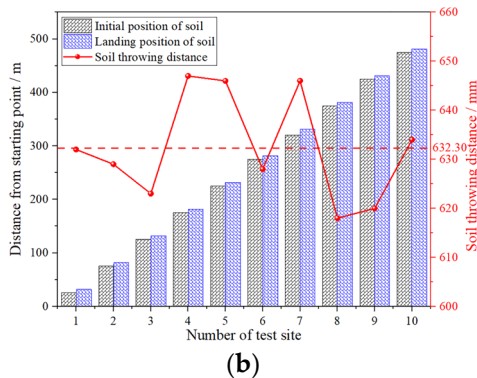

(**b**)

**Figure 21.** Soil throwing situation. (**a**) operation effect; (**b**) results of field experiment.

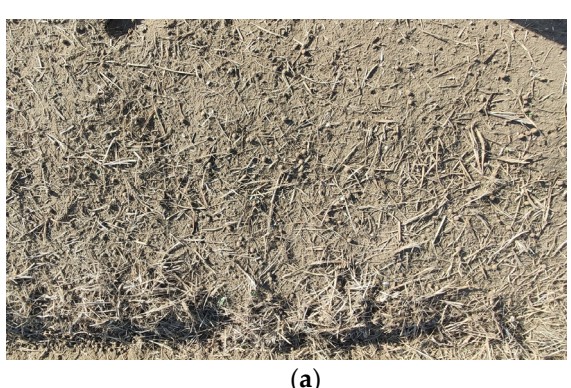

(**a**)

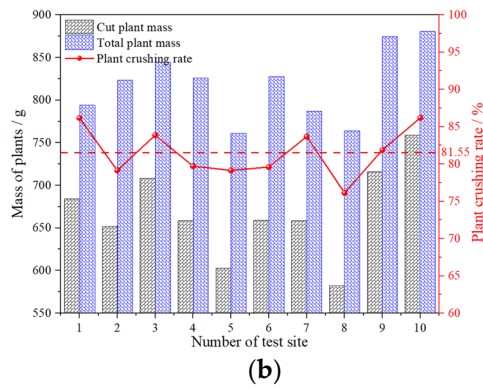

(**b**)

**Figure 22.** Plant crushing situation. (**a**) operation effect; (**b**) results of field experiment.

## 4. Discussion

At present, scholars have conducted research and achieved many results on the bionic optimization design of tillage machines and their soil-engaging components. Guan et al. [13] conducted bionic design for cutter teeth by studying the two-dimensional curve of badger feet. The field test results showed that the average power consumption of bionic cutter teeth was reduced by 0.01 kw. Qi et al. [9] studied and analyzed the three-dimensional tooth profile on the action pincers of crab claw foot, and used their findings to optimize the rotary blade edge. The results showed that, compared with the prototype rotary blade, the profiling rotary blade increased tillage depth by 2.6%, soil breaking rate by 2.3%, vegetation cover after tillage by 2.4%, and soil flatness after tillage by 26.7%. Among existing related research, one category was the two-dimensional projection of the main conformational features of the bionic object, and the other category was the simulation of biological three-dimensional configuration using inverse modeling. With increasing requirements for the performance, and the reduction of drag and consumption of tillage machinery, it is necessary to further enrich the bionic configuration design method.

In this study, the excavation edge and excavation surface of the rotary blade were bionic in three dimensions, and the prototype of the claw toe with the best operational performance was found by DEM. The operational performance of the bionic rotary blade, based on this claw toe, was compared with that of the rotary blades IT245 and IT245P [29]. As shown in Table 9, the production cost of the bionic rotary blade was slightly higher than that of rotary blades IT245 and IT245P. Compared to the rotary cutters IT245 and IT245P, the plant crushing rate of *Cyperus esculentus* of the bionic rotary blade increased by 35.45% and 27.79%, respectively, and the power consumption decreased by 14.62% and 3.45%, respectively. The results showed that the design ideas adopted in this study were feasible

and enriched the bionic conformal design method, which could provide bionic design ideas and methods for the design of soil cutting-based tillage machinery's soil-engaging components, such as the rotary blade and returning blade.

**Table 9.** Comparison of price and operational performance of different rotary blades.

|  | Rotary Blade IT245 | Rotary Blade IT245 | Bionic Rotary Blade |
|---|---|---|---|
| Cost (RMB) | 15 | 17 | 20 |
| Number of Broken Bonds | 1575 | 1762 | 2440 |
| Power Consumption/(kW) | 43.90 | 38.82 | 37.48 |

## 5. Conclusions

(1)　A bionic rotary blade was designed to address the problems of severe soil congestion and easy plant entanglement in the harvesting and excavation of *Cyperus esculentus*. Based on the unique combination of longitudinal soil cutting and lateral soil throwing of *Gryllotalpa orientalis Burmeister*, the edge curve and curved surface of the bionic rotary blade were designed using the excavation edge curve and excavation surface curvature of the toes of the mole cricket. Mechanical and kinematic analyses of the bionic rotary blade during the harvesting process of *Cyperus esculentus* were conducted. The parameters affecting the digging quality of *Cyperus esculentus* were determined to be the blade spacing, forward speed, and shaft speed.

(2)　A discrete element model of the flexible plant soil bionic rotary blade was established. A single factor experiment and a quadratic regression orthogonal rotational combination design were conducted, with the soil throwing distance and the number of broken bonds of *Cyperus esculentus* plants as evaluation indices. The results showed that the bionic rotary blade, designed with excavation edge and excavation surface based on the first claw toe of the mole cricket had the best operational performance. The best combination of the bionic rotary blade parameters were 11.16 mm for blade spacing, 0.66 m/s for forward speed, and 300 rpm for shaft speed.

(3)　The prototype was produced, and a field experiment was conducted according to the best parameters combination. The results showed that the bionic rotary blade's average soil throwing distance and plant crushing rate were 632.3 mm and 81.55%, respectively. The soil throwing and plant crushing performance of the bionic rotary blade were good, which could effectively solve the problems of severe soil congestion and low plant crushing rate. The operation performance of the bionic rotary blade was superior to that of the Chinese standard rotary blade IT245 and the rotary blade IT245P. The results of this study can enrich the bionic configuration design method and provide bionic design ideas and methods for the design of the soil-engaging components of tillage machinery, such as the rotary blade and the returning blade.

**Author Contributions:** Conceptualization, H.Z.; methodology, H.Z.; software, H.Z. and H.W.; validation, H.Z., D.W. and Z.Z.; formal analysis, H.Z. and Y.T.; investigation, H.Z.; resources, H.Z.; data curation, Y.S.; writing—original draft preparation, H.Z.; writing-review and editing, S.S. and D.W.; visualization, H.Z.; supervision, H.Z.; project administration, X.H. and D.W.; funding acquisition, X.H. and D.W. All authors have read and agreed to the published version of the manuscript.

**Funding:** This research was funded by Autonomous Region Science and Technology Support Project Plan (Grant NO.2020E02112) and Major Science and Technology Projects in Henan Province (Grant NO.211100110100).

**Institutional Review Board Statement:** Not applicable.

**Informed Consent Statement:** Not applicable.

**Data Availability Statement:** Not applicable.

**Conflicts of Interest:** The authors declare no conflict of interest.

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
