# Peer review of "Study on Plant Crushing and Soil Throwing Performance of Bionic Rotary Blades in Cyperus esculentus Harvesting"

_machines, doi:10.3390/machines10070562_

Round 1

Reviewer 1 Report

The work deals with quite interesting issues regarding the investigation of observed and analyzed the plant crushing and soil throwing performance of bionic rotary blades in Cyperus esculentus harvesting. However, revision is still needed before the acceptance of this manuscript.

Results in Table 1 and 2 are not described. It should be a discussion of the given results. Now, It is only indicated that the results are presented in the tables. The same remarks would apply to the other results in the tables (till Field Experiment).

Reviewer 2 Report

Correct the manuscript:

1. The relevance of the study is associated with an increase in the efficiency of the process of crushing plants of the Cyperus esculentus type and spreading the soil with rotating blades, and the novelty lies in the development of a rotating knife with a contour bend of the edge of the recess and the surface of the recess, similar to the clawed toe of the forefoot of Gryllotalpa orientalis Burmeister. The authors position the found design solution as a benchmark for improving the performance of such digging devices, but do not provide estimates of the reliability of the new device. It is important to assess how much the indicator of reliability, durability, maintainability has changed compared to the analogue. Now these calculations are not in the manuscript;

2. Annotation or keywords should contain information about EDEM software;

3. The interpretation of the regression equation (20) has not been fully carried out. It is necessary to explain the reasons for the influence of factors (mechanism);

4. Figures 19 and 20 should additionally indicate the value of the third fixed factor;

5. The method for assessing the degree of plant grinding should be explained (Fig. 22b). How significant is the subjective component in it?

6. In the conclusions and abstracts, add an assessment of the reliability of the new device and the prospect of developing your research.

Reviewer 3 Report

  1. It is necessary to present the physical and mechanical parameters of the soil interacting with the developed working body.
  2. It is necessary to present the values of energy costs for the implementation of the technological process of tillage of the developed machine.
  3. It is necessary to indicate the convergence of the results of theoretical and experimental studies to substantiate the parameters of the developed working body. 

Round 2

Reviewer 2 Report

Ok